# CCM-DiT: Camera-pose Controllable Method for DiT-based Video Generation

## Abstract

Despite the significant advancements made by Diffusion Transformer (DiT)-based methods in video generation, there remains a notable gap with camera-pose perspectives. Existing works such as OpenSora do not adhere precisely to anticipated trajectories, thereby limiting the utility in downstream applications such as content creation. Therefore, we introduce a novelty approach that achieves fine-grained control by embedding sparse camera-pose information into the temporal self-attention layers. We employ LoRA to minimize the impact on the original attention layer parameters during fine-tuning and enhance the supervision of camera-pose in the loss function. After fine-tuning the OpenSora's ST-DiT framework on the RealEstate10K dataset, experiments demonstrate that our method outperforms LDM-based methods for long video generation, while maintaining optimal performance in trajectory consistency and object consistency.

## 1 Introduction

The rapid evolution on video generation has been marked by the rise of DiT method Peebles & Xie (2023), which is well-suited for generating long video sequences. Despite these advances, DiT models often struggle with controllability, particularly in the precise control of camera movements, which is essential for many creative applications.

The recent video generation methods such as AnimateDiff Guo et al. (2023), Lumiere Bar-Tal et al. (2024), and SVD Blattmann et al. (2023a) have advanced from text-to-image (T2I) to text-to-video (T2V) domains by modifying the U-Net Ronneberger et al. (2015). Currently, the guidance by camera motion and object motion information, like MotionCtrl Wang et al. (2024b) and CameraCtrl He et al. (2024), takes more possibility to video content creation. However, these methods are mainly constrained by the Latent Diffusion Models (LDM) Rombach et al. (2022), which imposes strict limitations on the latent space, resulting in videos generated by U-Net fail to adjust resolution and duration. With the release of Sora Brooks et al. (2024) earlier this year, researchers start to focus on DiT-based methods. Recent works such as Kling, OpenSora Zheng et al. (2024), and Open-Sora-Plan Lab & etc. (2024) have conducted extensive explorations on 3D-VAE and spatial-temporal DiT (ST-DiT). These methods have achieved promising results in the T2V task. For applications involving motion manipulation, Tora Zhang et al. (2024b) has implemented the extraction of trajectory data into motion-guided fusion. However, there is currently no effective solution for the enhancement of controllable video generation with camera-pose sequences.

Therefore, we propose a **c**amera-pose **c**ontrollable **m**ethod for **DiT**-based video generation (**CCM-DiT**), which effectively embeds camera-pose sequences into DiT and generates videos according to the corresponding camera-pose sequence. Our method utilizes the OpenSora-v1.2 framework and extracts inter-frame motion sequences from reference videos in camera perspectives. First, each frame is annotated with a 12-dimension motion matrix, including a $3 \times 3$ rotation matrix and a $3 \times 1$ translation matrix. Effectively capturing the precision of camera-pose remains a challenge. We propose the Sparse Motion Encoding Module for converting a pixel-wise motion field based on Plücker coordinates into a sparse motion field. Second, compared to the U-Net, the DiT framework compresses the temporal dimension to reduce VRAM usage, making it difficult to align frame-based motion information with the temporal attention layer, thus complicating the embedding of camera-pose motion. Inspired by Tora, we train a VAE Kingma (2013) for the latent space of camera-pose sequences, improving its alignment with the temporal attention layer.

Figure 1: The overview of the CCM-DiT. CCM-DiT includes the Sparse Motion Encoding Module and the Temporal Attention Injection Module. It establishes a sparse motion sequence representation based on Plücker coordinates and feeds it into the VAE for latent space encoding, handling the camera-pose sequences for multiple frames. By employing the adaptive normalization method, it achieves alignment of the temporal attention layer and the latent motion. The inputs of the video and text caption are consistent with OpenSora, feeding into the ST-DiT and cross-attention layers through the 3D-VAE and T5 model, respectively.

The training of CCM-DiT consists two parts. First, the reconstruction loss is used for the camera-pose sequences during VAE training. We use RealEstate10K Zhou et al. (2018), a video dataset with over 60k camera-pose annotations, to train the VAE for sparse motion sequences. Second, we fine-tune the OpenSora by introducing a motion injection module after the VAE. To reduce memory usage, most of the original parameters are frozen while applying LoRA in the temporal attention layer. We evaluate our method and the experiments show that our approach achieved state-of-the-art (SOTA) performance for long video generation tasks.

Our main contributions are:

- We propose a method to embed camera-pose sequences into the DiT framework, enabling video generation to accurately follow camera-pose motion.

- We introduce sparse motion encoding module and LoRA fine-tuning for temporal attention, allowing for efficient encoding of camera-pose sequences. Meanwhile, we design a loss function related to camera-pose.

- Our method achieves SOTA during long video generation with camera-pose sequences.

## 2 RELATED WORK

### 2.1 VIDEO GENERATION

With diffusion models being proven as an effective method for creating high-quality images, research on dynamic video generation has gradually emerged. Make-a-video Singer et al. (2022) and MagicVideo Zhou et al. (2022) use 3D U-Net in LDM to learn temporal and spatial attention, though the training cost is relatively expensive. VideoComposer Wang et al. (2024a) expands the conditional input forms by training a unified encoder. Other methods (Align Your Latents Blattmann et al. (2023b), VideoElevator Zhang et al. (2024a), AnimateDiff, Direct a Video Yang et al. (2024a), Motioni2v Shi et al. (2024), Consisti2v Ren et al. (2024)) improve the performance by reusing T2I models and make adjustments in the temporal and spatial attention parts to reduce issues such as flicker reduction. Video generation models based on DiT or Transformer Vaswani (2017) adopt spatial-temporal attention from LDM, such as Latte Ma et al. (2024), Vidu Bao et al. (2024), CogVideoX Yang et al. (2024b) and SnapVideo Menapace et al. (2024), which have significant advantages in terms of resolution and duration compared to LDM methods.

### 2.2 CONTROLLABLE GENERATION

Controllable generation is one of the key research topics for generative tasks. For T2I task, ControlNet Zhang et al. (2023) enables fine-tuning samples while retaining the backbone, and Control-

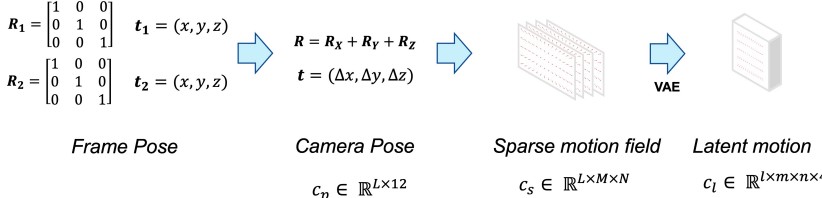

Figure 2: VAE to encode camera-pose sequences. The matrix parameters between adjacent frames are calculated to obtain the camera-pose sequence, which is then transformed into RGB space through the sparse motion field and finally processed into motion latent by the VAE.

NeXT Peng et al. (2024) significantly improves training efficiency. For controllable video generation, tune-a-video Wu et al. (2023) enables single sample fine-tuning, changing styles while maintaining consistent object motion. MotionClone Ling et al. (2024) implements a plug-and-play motion-guided model. MotionCtrl and CameraCtrl use motion consistency modules to introduce camera-pose sequences. PixelDance Zeng et al. (2024) uses the first and the last frame as reference for video generation. Image Conductor Li et al. (2024) and FreeTraj Qiu et al. (2024) introduce tracking schemes based on trajectories and bounding boxes, respectively. ViewDiff Höllein et al. (2024) reconstructs 3D information of objects based on camera-pose sequences. As for Transformer or DiT, there are few researches for camera-pose. VD3D builds on SnapVideo, embedding camera-pose into cross-attention layers via Plücker coordinates. Tora and TrackGoZhou et al. (2024) explore controllable video generation by trajectories and masks. Currently, there is still limited work for camera-pose information on DiT.

## 3 METHOD

### 3.1 PRELIMINARY

The LVDM (Latent Video Diffusion Model) He et al. (2022) aims to video generation through a denoising diffusion network like U-Net. It proposes a strategy for the separation of spatiotemporal self-attention to address the frame motion coherence in video generation. The loss function for the U-Net is shown in the following formula:

$$\mathcal{L}(\theta) = \mathbb{E}_{z_0,c,t,\epsilon}[\|\epsilon_\theta(z_t, c, t) - \epsilon\|_2^2] \tag{1}$$

Here, $\epsilon_\theta$ is the predicted noise, $z_t$ and cc represent the latent space at $t$ step and text condition, respectively. The latent space of the U-Net conforms to the following Markov chain:

$$z_t = \sqrt{\bar{\alpha}_t} z_0 + \sqrt{1 - \bar{\alpha}_t} \epsilon \tag{2}$$

where $\bar{\alpha}_t = \prod_{i=1}^{t} \alpha_t$, $\alpha_t$ represents the noise strength in step $t$.

The DiT-based method replaces the U-Net with Transformer, remaining its sequential processing capabilities to greatly enhance the image quality and duration in video generation. To reduce computational complexity, the 3D-VAE in OpenSora performs a $4\times$ compression on the temporal dimension. Compared to LVDM's latent space of $b \times L \times w \times h$, OpenSora's latent space size is $b \times f \times w \times h (f = L/4)$, which is more lightweight on the temporal dimension.

### 3.2 CCM-DiT

As depicted in Fig. 1, the proposed CCM-DiT consists of two modules: the Sparse Motion Encoding Module and the Temporal Attention Injection Module. Previous works describe camera motion in various ways, such as using Plücker coordinates He et al. (2024); Bahmani et al. (2024) or directly based on motion matrices Wang et al. (2024b). Methods based on Plücker coordinates calculate the Plücker embedding for each pixel in the image coordinate space, with the corresponding equation:

$$\mathbf{P}_{x,y} = [\mathbf{o}_c, 1] \left( \mathbf{RK}^{-1} [x, y, 1]^T + \mathbf{t} \right) \tag{3}$$

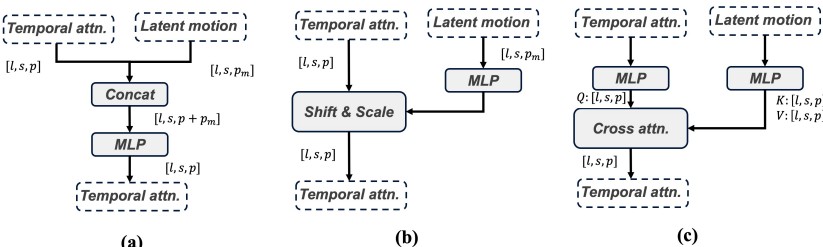

Figure 3: Embedding Methods. (a) channel-dimension concatation directly; (b) adaptive normalization, the latent motion is scaled and shifted for alignment with temporal attention; and (c) cross-attention, while temporal attention and latent motion are reconstructed for cross-attention.

Here, $\mathbf{o}_c, \mathbf{t} \in \mathbb{R}^{3 \times 1}$ represent the camera center and the translation part, and $\mathbf{R}, \mathbf{K}$ are the rotation matrix and intrinsic parameters of the camera-pose. $\mathbf{R}\mathbf{K}^{-1}[x, y, 1]^T + \mathbf{t}$ forms the direction vector from the camera center to the pixel point $(x, y)$. For the method of directly using motion matrices, camera poses are serialized frame-by-frame into $\mathbf{RT} \in \mathbb{R}^{L \times 12}$, where $L$ denotes the frame number. During motion injection, the parameters are replicated in spatial dimension to align temporal attention layer. However, this approach may encounter problems for DiT-based method that exists time-dimensional compression.

**Sparse Motion Encoding Module.** In this work, we propose a method for converting a pixel-wise motion field based on Plücker coordinates into a sparse motion field, as shown in Fig. 2. Although Plücker coordinates can precisely describe the motion trajectory for each pixel in the image, we perform sparse sampling of the motion field to enhance computational efficiency and adapt to spatial domain feature representations. Specifically, the image is sampled at regular intervals and Plücker motion vectors are calculated on these sparse points, forming a sparse motion vector field. Assuming the image resolution is $W \times H$, we sample every $s_x$ pixels in the $x$ direction and every $s_y$ pixels in the $y$ direction to obtain a sparse point sequence $\{(x_i, y_j)\}$, with the corresponding sparse motion trajectory given by:

$$\mathbf{P}_{x_i, y_j} = [\mathbf{o}_c, 1] \left( \mathbf{R}\mathbf{K}^{-1}[x_i, y_j, 1]^T + \mathbf{t} \right) \tag{4}$$

where $x_i = i \cdot s_x$ and $y_j = j \cdot s_y$, with $i$ and $j$ being the sampling indices. Here, we get a sparse motion field $F_s \in \mathbb{R}^{L \times M \times N}$, the $M = W/s_x$, $N = H/s_y$.

We trained a VAE to compress the sparse motion field, aligning it with the temporal sequences in OpenSora. MegViT-v2 Yu et al. (2023) is selected to maintain consistency with the temporal attention layers and the reconstruction loss of the camera-pose motion is calculated.

**Temporal Attention Injection Module.** We consider three typical embedding methods, including channel-dimension concatation, adaptive normalization, and cross-attention, as shown in Fig. 3. Direct channel-dimension concatation adds the camera-pose motion latent to the temporal layers, which is used in MotionCtrl. Adaptive normalization uses multi-layer perceptron (MLP) for latent motion alignment with temporal layers. $\beta, \gamma$ are used for shift & scale during linear projection, respectively. For cross attention, temporal layers represents query, while latent motion represents key and value, calculates the hidden layers. We experiment three injection methods, and adaptive normalization gives the best performance and consistency during video generation.

### 3.3 TRAINING DETAILS AND DATA PROCESSING

**Training Details.** The Open-Sora's second training stage is ultilized to train the VAE of camera-pose sequences. Specifically, the training strategy supervises the reconstruction process, including reconstruction loss and KL loss. The reconstruction loss aims to minimize the gap between the predicted result and the ground truth, while the KL loss minimizes the divergence between the VAE's output distribution and the standard normal distribution. During the fine-tuning of the latent motion using MLP, we freeze the ST-DiT parts except for the temporal attention layer, and introduce LoRA during the update of the self-attention to reduce VRAM usage. Additionally, a novelty loss

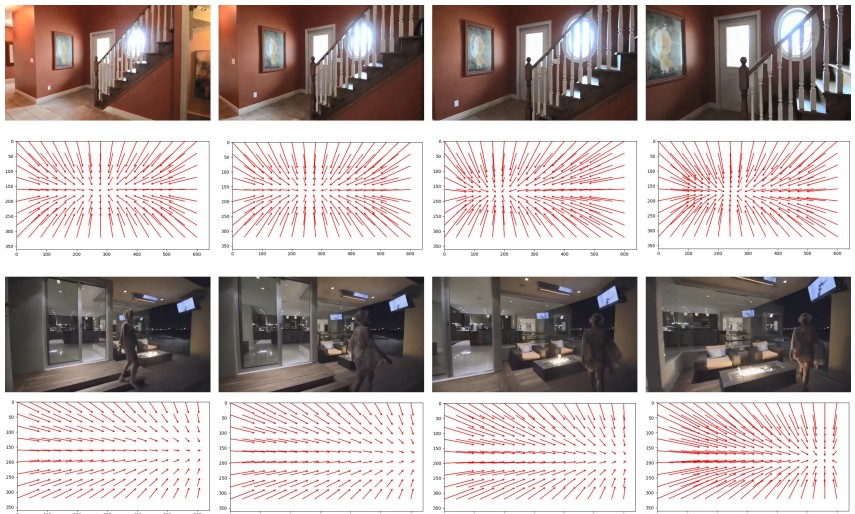

Figure 4: Camera-pose visualization. We visualize image sequence after sparse motion sampling, with each row representing frame 0, frame 5, frame 10, and frame 15 (final frame) of the camera-pose series from left to right. The arrows in the image indicate the motion of the sampling points. The first row shows a camera zoom-in motion, and the second row shows a pan-right motion.

function is introduced for fine-tuning, which incorporates $p_m$ as camera-pose motion conditional inputs, comparing to equation 1.

$$\mathcal{L}(\theta) = \mathbb{E}_{z_0, c, t, \epsilon, p_m}[\|\epsilon_\theta(z_t, c, t, p_m) - \epsilon\|_2^2] \tag{5}$$

**Data Processing.** Various forms of condition input, including camera-pose representation, text prompt and reference image, are carefully considered before fine-tuning. For a better camera-pose representation, we randomly select 17-frame video segments and get their 12-point camera-pose from timestamp information. Then we use sparse motion sampling method mentioned in Section 3.2 to get the RGB image of the motion field as the camera-pose representation, which gets the alignment with the sampling frame motion. For text prompt and reference image, we follow the pretrained model in OpenSora, with T5 model and 3D-VAE model, respectively.

## 4    EXPERIMENTS

### 4.1    IMPLEMENTATION DETAILS

We use the weights and network structure following OpenSora-v1.2. When training the Sparse Motion Encoding, only the parameters of the motion-relative part and the temporal-attention part are trained, while the backbone parameters are frozen to retain the original capabilities. During training, the approach of MotionCtrl is followed. We extract 16-frame camera pose information and convert it into a RGB sparse representation (as shown in Fig. 4), and feeding it into the VAE for reconstruction. The guidance scale is set to 7.0. We fine-tunes on 4 Nvidia L40s with the learning rate of $5 \times 10^{-5}$, requiring 100k steps and with the batch size of 1, which takes approximately 2.5 days.

### 4.2    DATASETS

To validate the effectiveness of the proposed method, we use the RealEstate10K dataset, consistent with MotionCtrl and VD3D. We randomly select 20 videos from the test set, which include common camera movements such as pan left/right, up and down, zoom in/out, as well as roundabout and other complex movements.

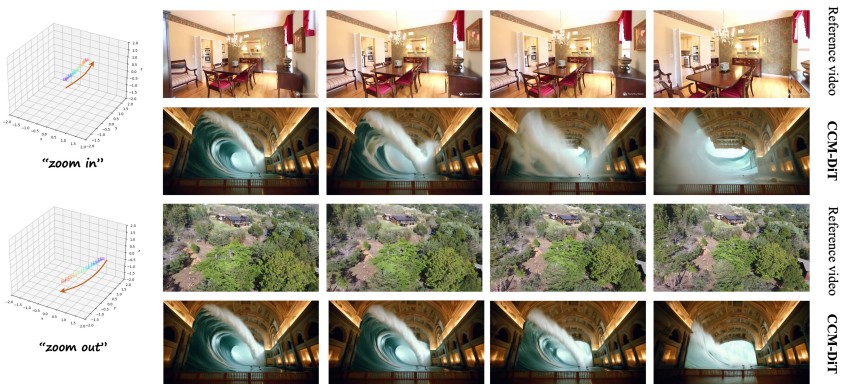

Figure 5: The video generation performance on basic camera movements. The text prompt is: The waves are surging inside the house. Each row representing frame 0, frame 23, frame 47 and frame 71 (last frame) of the actual video. The first row shows a camera zoom-in motion, and the second row shows a camera zoom-out motion.

### 4.3 METRICS

We use Fréchet Inception Distance (FID)Heusel et al. (2017), Fréchet Video Distance (FVD) Unterthiner et al. (2018), and CLIP Similarity (CLIPSIM) Radford et al. (2021) as metrics to evaluate the image quality, video consistency, and semantic similarity of the generated videos. For the camera-pose consistency metric, we adopt the CamMC, the same approach mentioned in MotionCtrl. Since DiT demonstrates advantages in long video generation, we test the performance of video generation extended to 72 frames. For LDM methods, we produce long videos by using the final frame of the previous segment as the reference for the subsequent segment.

### 4.4 QUANTITATIVE AND QUALITATIVE RESULTS

We evaluate the performance of several video generation models on both short video (16 frames) and long video (72 frames) generation tasks. The methods include LDM-based approaches such as SVD, AnimateDiff, MotionCtrl, and CameraCtrl, and DiT-based methods like EasyAnimate, VD3D, and OpenSora, as shown in Table 1. The resolution for LDM-based methods is mainly $256 \times 256$ or $384 \times 256$, while DiT-based methods use a unified resolution of $640 \times 360$. For short video generation tasks, MotionCtrl shows an advantage, achieving the best results in video consistency metrics (FVD and CamMC). However, in long video generation tasks, CCM-DiT demonstrates significant advantages in consistency metrics. This is mainly attributed to CCM-DiT's more precise camera-pose sequences input during long video generation, which allows for fine-grained control over each frame. Additionally, it outperforms previously proposed methods in the CLIPSIM metric as well, which demonstrates that CCM-DiT effectively retains reference image. This is because we freeze other irrelevant parameters as much as possible when introducing camera-pose sequences, preserving the model's original video generation capabilities.

We also present the visualized performance of video generation using CCM-DiT (Fig. 5 and 6). For simple camera-pose, such as zoom in and zoom out, CCM-DiT performs excellently on these basic camera movement tasks, accurately following the camera motion poses. For complex tasks, such as camera movement with rotation, CCM-DiT achieves smooth transitions while maintaining the object pose effectively.

### 4.5 ABLATION STUDIES

We conduct ablation studies for CCM-DiT, focusing on the sampling interval of camera-pose RGB series and the temporal injection methods, corresponding to the Sparse Motion Encoding and Temporal Attention Injection Module introduced in Section 3.2.

In the sampling interval experiment, we conduct three sets of motion extraction strategies: $20\times$, $40\times$, and $80\times$. For example, for $640 \times 360$ video resolution, the $40\times$ strategy corresponds to $16 \times 9$

Figure 6: The video generation performance on complex camera movements. The text prompt is: The dog is watching and moving around. Each row representing frame 0, frame 23, frame 47 and frame 71 (last frame) of the actual video. This case shows a camera roundabout motion.

Table 1: Comparison of consistency performance using different video generation methods, our method CCM-DiT achieves the best results in long video task.

| Models | FID (↓) | | FVD (↓) | | CLIPSIM (↑) | | CamMC (↓) | |
|---|---|---|---|---|---|---|---|---|
| | Short | Long | Short | Long | Short | Long | Short | Long |
| SVD Blattmann et al. (2023a) | 185 | 261 | 1503 | 1628 | 0.1604 | 0.1102 | 0.160 | 0.885 |
| AnimateDiff Guo et al. (2023) | 167 | 175 | 1447 | 1512 | 0.2367 | 0.2045 | 0.051 | 0.473 |
| MotionCtrl Wang et al. (2024b) | **132** | 168 | **1004** | 1464 | 0.2355 | 0.2268 | **0.029** | 0.472 |
| CameraCtrl He et al. (2024) | 173 | 254 | 1426 | 1530 | 0.2201 | 0.2194 | 0.052 | 0.205 |
| EasyAnimateV3Xu et al. (2024) | 165 | 245 | 1401 | 1498 | 0.2305 | 0.2250 | 0.046 | 0.068 |
| VD3DBahmani et al. (2024) | – | 171 | – | 1400 | – | 0.2032 | – | 0.044 |
| OpenSora Zheng et al. (2024) | 141 | 161 | 1587 | 1682 | 0.2496 | 0.2284 | – | – |
| CCM-DiT (Ours) | 147 | **158** | 1310 | **1387** | **0.2521** | **0.2438** | 0.037 | **0.042** |

Table 2: Ablation study results showing the effect of sample ratios for camera pose latents.

| Ratios | FID (↓) | FVD (↓) | CLIPSIM (↑) | CamMC (↓) |
|---|---|---|---|---|
| 20× | 156 | 1395 | 0.2328 | 0.045 |
| 40× | **148** | **1313** | **0.2521** | **0.038** |
| 80× | 151 | 1358 | 0.2462 | 0.042 |

motion extraction points. We train the VAE using different sampling strategies and evaluate the video generation performance, as shown in Table 2. We find that the 40× achieves the best results across all metrics, indicating that the camera-pose motion sampling quantity at 40× is relatively optimal. For the 20× and 80×, we observe varying degrees of target drift or weakened motion consistency during evaluation. The possible reason is that for 80×, the sampling density is sparse (around 40 vectors per frame), making it easy for targets to be distorted and reducing motion control capability. On the other hand, for 20×, there are over 500 vectors each frame, making it difficult to align with each motion vector and leading to a decrease in motion consistency. This ablation study provides a reference for quantifying sparse motion sampling.

In the injection method experiment, we also use three strategies: channel-dimension concatenation (concat), adaptive normalization, and cross-attention. The video generation performance for the three methods are shown in Table 3. We find that adaptive normalization achieves better consistency results compared to the other methods. The reason is that for channel-dimension concatenation, which fails to align the motion latent with the temporal attention at first, leading to weaker camera-pose control during generation. For cross-attention, which alters the dimension of both motion latent and temporal attention, causes more disruption to the temporal attention in the original network. Additionally, we observe that adaptive normalization is able to unify motion and temporal latent into a similar distribution, which is crucial for the effective injection of camera-pose.

Table 3: Ablation study results showing the effect of different injection modules for camera-pose latents.

| Methods | FID (↓) | FVD (↓) | CLIPSIM (↑) | CamMC (↓) |
|---------|---------|---------|-------------|-----------|
| Concat | 152 | 1342 | 0.2328 | 0.046 |
| Cross Attn. | 149 | 1326 | 0.2335 | 0.041 |
| Adapt. Norm. | **148** | **1313** | **0.2521** | **0.038** |

## 4.6 DISCUSSIONS

CCM-DiT demonstrates excellent performance in maintaining camera consistency for long video generation, but there are still the following challenges and limitations:

- **The performance of the main object is relatively weak.** We focus on maintaining the consistency of camera-pose motion. Although object consistency is also preserved, due to the conservative nature of motion estimation, the object movement tends to be limited to small-scale motions, making large-scale motion generation more challenging.

- **There is limited support for camera-pose motion trajectories.** To ensure consistency in our study, we use camera-pose condition based on 16 frames. More frame requirements rely on frame interpolation for completion. Currently, generating more complex motion videos remains a challenge.

## 5 CONCLUSION

We propose a novelty method for camera-pose controllable video generation based on DiT architecture. To effectively inject camera-pose sequences into the temporal-attention layer, we introduce a Sparse Motion Encoding Module that transforms motion into sampling points in the RGB space and use a VAE to achieve latent motion feature embedding. Our method achieves SOTA in camera motion control for long video scenarios. We believe this work will find valuable applications in the future of controllable video creation.

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
