# OpenReview forum: "CCM-DiT: Camera-pose Controllable Method for DiT-based Video Generation"
_ICLR.cc/2025/Conference — ICLR 2025 Conference Withdrawn Submission_

### Official Review · Reviewer_pqnn · 2024-10-27

**Soundness:** 2
**Presentation:** 1
**Contribution:** 2
**Rating:** 3
**Confidence:** 3

**Summary:**

In this paper, the authors propose a DiT-based video generation method that embeds the camera poses as controllable signals. They use LoRA to fine-tune the attention layer parameters in the training. RealEstate10K dataset is used in the evaluation.

**Strengths:**

1. The authors add the camera pose in the video generation, which is an interesting point.

**Weaknesses:**

1. Many details are missing in Fig 2. I do not find any explanations of what does frame pose mean and how to get the frame pose. How to convert frame pose to camera pose is also very unclear. It would be helpful to provide a step-by-step description of how to extract and process the pose information from the dataset, including definitions of frame pose and camera pose, and the conversion process between them.
2. The method is only evaluated on a single dataset, which is not sufficient to verify the effectiveness of the method. For example, authors can test on videos from WebVid and HD-VILA following MotionCtrl [1] paper.

[1] Zhouxia Wang, Ziyang Yuan, Xintao Wang, Yaowei Li, Tianshui Chen, Menghan Xia, Ping Luo, and Ying Shan. 2024. MotionCtrl: A Unified and Flexible Motion Controller for Video Generation. In ACM SIGGRAPH 2024 Conference Papers (SIGGRAPH '24). Association for Computing Machinery, New York, NY, USA, Article 114, 1–11. https://doi.org/10.1145/3641519.3657518

**Questions:**

1. In Fig 1, I am curious how to convert the text instruction "Zoom-in" to the motion field.
2. Eq (3) is unclear. Is P_{x,y} the  Plucker embedding? But how to get R, K and t from the dataset? Does the dataset provide such information?
3. What is the major difference between this paper and the motionCtrl [1]? A detailed comparison of the proposed method with MotionCtrl, highlighting key differences in approach, architecture, and performance, would be helpful.

[1] Zhouxia Wang, Ziyang Yuan, Xintao Wang, Yaowei Li, Tianshui Chen, Menghan Xia, Ping Luo, and Ying Shan. 2024. MotionCtrl: A Unified and Flexible Motion Controller for Video Generation. In ACM SIGGRAPH 2024 Conference Papers (SIGGRAPH '24). Association for Computing Machinery, New York, NY, USA, Article 114, 1–11. https://doi.org/10.1145/3641519.3657518

---

### Official Review · Reviewer_6DBB · 2024-10-29

**Soundness:** 3
**Presentation:** 2
**Contribution:** 2
**Rating:** 3
**Confidence:** 5

**Summary:**

This paper aims at controlling the camera viewpoints of the videos generated by the DiT-based video diffusion models. To achieve the precise camera viewpoint control, this paper utilizes the Plücker embedding as the camera representation. The Plücker embeddings are per-frame spatial maps, while the DiT-based diffusion (like OpenSora) do some downsamples in the temporal dimension. To deal with this conflict, this paper proposes a Sparse Motion Encoding Module to temporally downsample the Plücker embeddings, with the same ratio as the OpenSora VAE. This Sparse Motion Encoding Module is implemented by a MagVit2 like causal VAE. The generated latent motion is injected into the temporal attention layer of OpenSora using an adaptive normalization layer. Experiments demonstrate the superiority of proposed method on both short and long video generation (with camera control) task. Some ablation studies also prove the effectiveness of the proposed Sparse Motion Encoding Module and Temporal Attention Injection Module.

**Strengths:**

1. This paper train a VAE model to temporally downsample the Plücker embeddings, easing the conflict of different temporal length between Plücker embedding and the latent features.
2. The visualization results demonstrate the effectiveness of the proposed method on some simple camera trajectories, like paning and zooming.

**Weaknesses:**

1. The motivation of Sparse Motion Encoding Module is not well presented. Why the VAE model is used to compress the Plücker embedding? The encoder of VAE can be used to compress the Plücker embedding, what is the decoder used for? Besides, generally, the VAE model will not bring too much extra computation, and it can be reused once encoded, why this module **sparsely** sample some Plücker motion vector?
2. The writing is not good for this manuscript. For example, there are some typos, like the **MegVit-v2** in Line 198, the inconsistency of OpenSora and Open-Sora. Besides, some details are missing, like what is the input of the Sparse Motion Encoding Module. The rows 2 and 4 in Figure 4 does not provide too much information.
3. The experiments is not very convincing. See Question part.

**Questions:**

Besides the questions in the first point of weakness section, I have the following questions.
1. The MagVit2 model is designed to be able to compress the images and videos in a single model. Using some padding, the first image is treated as a separate image, thus the training of MagVit2 VAE is 17 frames (line 17) is reasonable. But, in line 259, the author said "we extract 16-frames...", I want to know how those 16 frame are padded and what is the output of the VAE encoder?
2. Can the author provide the reconstruction results, using l1 loss, for the reconstructed sparse Plücker embedding?
3. Whether the motion degree of objects of the whole scene degraded after adding the camera control-related modules?
4. In line 303, the author state that they use the different resolution for different video generation models, can the FID, FVD fully reflect the ranking of different models?
5. In the visualization results, the camera trajectories seems too simple, focusing on panning and zooming. I remember there are some more complex camera trajectories in RealEstate10K dataset, can the author provide some quantitative or qualitative results on those complex camera trajectories?
6. How to calculate the CamMC metric for SVD, AnimateDiff, OpenSora model, since they cannot take the camera trajectories as input.

---

### Official Review · Reviewer_J1Rw · 2024-11-02

**Soundness:** 2
**Presentation:** 2
**Contribution:** 2
**Rating:** 3
**Confidence:** 3

**Summary:**

This paper presents an approach to training a DiT-based video diffusion model (OpenSora) to control the camera motions of generated videos.
The camera motion, which is represented by Plücker coordinates, is first sparsely sampled (downsampled by x40) and then encoded into the "motion latent" via VAE encoder (MegViT-v2, inspired by Tora). Finally, motion latent is injected into the temporal attention layer of DiT via adaptive normalization. The model is finetuned on 16-frame videos from the RealEstate10K dataset. The authors demonstrate that visual quality and motion accuracy (FID, FVD, CamMC) outperformed baselines for 72 frame generation.

**Strengths:**

- To the best of my knowledge, this is the first work that tackles camera-motion-controlled video generation with open-source DiT (i.e. opensora).

- The idea of sparsely sampling motion fields before inputting them into the VAE encoder is new.

- They demonstrate that adaptive normalization for conditioning camera motions is the effective strategy for camera-conditioned video generation for the first time, which is consistent with the results demonstrated in trajectory-controlled generation (e.g. Tora).

- They quantitatively demonstrate that the generated videos have better motion and visual quality for 72-frame generation.

**Weaknesses:**

Novelty:

- As the authors have already acknowledged, the idea of using Plucker coordinates has already been introduced (e.g., VD3D). Additionally, the use of a VAE encoder and adaptive normalization has already been introduced by Tora. Following that, the main technical contribution is introducing a sparsely sampled motion field. The author argues that sparsely sampled motion fields contribute to performance improvement, but the authors fail to provide details results (e.g., visual results, more ablation study in Table 2, what about x1?) nor detailed motivation. Additionally, choosing this downsample factor seems heuristic with no intuitive justifications. I would appreciate it if the authors could provide more ablation studies and technical motivations for applying sparsely sampled motion fields.

Experiments:

- Although the model is trained on a 16-frame dataset, the model performs worse than other baselines for 16-frame generation. Additionally, the motivation for training only on 16-frame videos is unclear, given the model is tasked to generate longer-frame videos during the experiments. I would appreciate it if the authors could provide more explanations for this decision.

- The authors did not provide sufficient qualitative results or user study, where the superiority of their method is not convincing.

Clarity:

- The paper lacks implementation details. For instance, I am not sure how adaptive normalization and cross-attention are performed exactly. Figure 3 seems inaccurate because the injection happens between two temporal attention layers, where the temporal attention layer should exist in only one of these locations. Please see the questions below for more requests for clarification.

**Questions:**

- Could the authors provide a user study to assess the quality?

- Could the authors upload the generated videos and visual comparisons with baselines?

- Could the authors provide the implementation details of cross-attention and adaptive normalization? Including where these computations happen in relation to temporal attention computation. (In Figure 3, the injection happens between two temporal attention layers. This figure is wrong to me as exactly one temporal attention should exist either before or after the injection.)

- Could the authors provide the reasons why the models are trained on only 16-frame videos?

- Could the authors detail how the model is extended to 72 frames given the model is trained on 16 frames?

- The authors mention that "object movement tends to be limited to small-scale motions". I think this can be a big issue. Could the authors provide a detailed comparison with other baselines?

- Is the motivation for introducing sparse sampling of the motion field for computational efficiency? The authors lately argue that sparse sampling improves the results, but I am not convinced why the performance is improved. Could the authors provide more detailed reasons behind that?

---

### Official Review · Reviewer_g6Tj · 2024-11-05

**Soundness:** 2
**Presentation:** 1
**Contribution:** 1
**Rating:** 3
**Confidence:** 4

**Summary:**

This paper presents an approach to enable controlling camera pose for video generation based on Diffusion Transformer (DiT). It converts pixel-wise motion field based on Plucker coordinates into a sparse motion field, which is then injected into the temporal attention part of DiT. LoRA is used to fine-tune a pre-trained DiT model (Open-Sora). Experimental results on the RealEstate10K dataset are reported.

**Strengths:**

1. The idea of converting camera poses into pixel-wise embeddings is novel, which allow the video generation model to effectively understand the camera motion.

2. This paper studies three common ways of incorporating camera pose embedding into a DiT model, which could be useful for future work.

**Weaknesses:**

1. The proposed sparse motion encoding module seems almost identical to the standard pixel-wise Plucker embedding. Compared with Eq. (3), the only difference in Eq. (4) is the embedding computation are performed on a set of sparse locations controlled by $s_x$ and $s_y$. It is not clear how the camera "motion" is encoded. Does the proposed approach convert the pixel-wise motion vectors shown in Fig. 4 into embeddings?

2. A lot of definitions are not clear and math symbols are not used rigorously in the presentation, making the paper hard to follow and understand.

- a) The Sparse Motion Encoding Module and Temporal Attention Injection Module are not shown in Fig. 1 at all.

- b) In line #179 on page 4, how is $RT$ defined? Is it matrix multiplication similar to $RK^{-1}$?

- c) In line #195 on page 4, it says $F_s\in \mathbb{R}^{L\times M\times N}$? According to the definition in Eq.(4), the channel dimension shouldn't be 1?

- d) In Fig. 2, $c_p$, $c_s$, and $c_l$ are not defined in the main text. And the shape of $c_l$ is not clearly explained either.

- e) In Fig. 3, what are $s$, $p$, and $p_m$?

3. The first two items of the claim contributions of the paper are essentially identical. Both of them are about incorporating camera poses into a DiT model.

4. Only visual results of simple camera motion (zoom in, zoom out, and roundabout) are shown in the paper. No supplementary results are available. It is therefore hard to gauge the effectiveness of the proposed approach.

**Questions:**

N/A

---

### Note · Authors · 2024-11-14

**Comment:**

Unauthorized submission of manuscripts without the consent of co-authors.

**Withdrawal Confirmation:**

I have read and agree with the venue's withdrawal policy on behalf of myself and my co-authors.